# MatchingPolicy: Correspondence-Aware Policy for Cross-Object In-Context Learning

## Abstract

In-context imitation learning allows policies to generalize from few-shot demonstrations, yet it often struggles with unseen objects or novel scenarios. To address this, we introduce MatchingPolicy, a correspondence-driven framework that decouples demonstration-to-scene matching from policy learning. At its core, MatchingPolicy uses a graph-based diffusion policy that adapts robot actions via dense semantic correspondences. This explicit separation eliminates the challenge of simultaneous correspondence inference and action adaptation, enabling robust transfer. Our approach integrates a vision foundation model with a novel two-stage matching algorithm to dynamically establish reliable correspondences. Empirical evaluations on RLBench and real-world manipulation tasks show that MatchingPolicy achieves strong few-shot performance, generalizing consistently across unseen object instances and categories. Visual results are available on our website matchingpolicy.github.io

## 1 Introduction

Imitation learning has demonstrated remarkable potential in acquiring diverse manipulation skills. However, traditional approaches typically require large amounts of costly, task-specific demonstration data to train a policy for even a single skill. While recent efforts have sought to train universal, multi-task imitation policies (Kim et al., 2024) on large and diverse datasets (O'Neill et al., 2024), these models still struggle to generalize to entirely new tasks and often depend on task-specific fine-tuning to reach acceptable performance. Inspired by breakthroughs in language and vision, in-context learning has recently been introduced to robotic policy learning (Fu et al., 2024) as a way to bypass fine-tuning. In this paradigm, a model is conditioned at test time on a few task demonstrations that provide fine-grained, step-by-step guidance—rather than coarse natural language commands—enabling the policy to adapt to unseen tasks on the fly.

Despite this promise, prior in-context learning approaches for robotics often fail to generalize to novel objects and tasks, even when the demonstrations feature the exact unseen task and object (Jain et al., 2024). We argue that a key reason for this shortcoming is that current methods ask a single model to simultaneously: (1) infer precise correspondences between the demonstration and the current scene from scratch using limited robotic-domain data, and (2) adapt the demonstrated actions to the new scene. This dual burden can hinder both correspondence extraction and action adaptation, especially in visually diverse or geometrically complex environments.

To address this challenge, we propose MatchingPolicy, a novel in-context imitation learning framework for robotic manipulation that explicitly decouples correspondence extraction from policy learning. Instead of forcing the policy to discover correspondences implicitly, we focus on training a policy that adapts robot behavior based on pre-computed correspondences. In addition, we introduce an online adaptive semantic matching algorithm that dynamically establishes dense and reliable correspondences between the demonstrations and the current scene.

Our correspondence-aware policy offers three key advantages. First, by leveraging explicit correspondences between the demonstration and the current scene, the policy can more precisely localize task-relevant interaction regions, enabling more accurate and adaptive action generation. Second, because the policy operates on correspondence features rather than raw visual object representations, it becomes inherently object-agnostic, substantially enhancing its ability to generalize across different object instances and categories. Third, the decoupled design—separating correspondence

extraction from policy learning—facilitates the seamless integration of powerful off-the-shelf visual and semantic matching methods, such as DINO (Oquab et al., 2023) and SD-DINO (Zhang et al., 2023), without retraining the policy network.

Our results show that the proposed MatchingPolicy outperforms the state-of-the-art in-context imitation learning method on the RLBench benchmark. We further validate our approach on a set of challenging real-world manipulation tasks, where it demonstrates strong generalization to a wide variety of objects. The main contributions of this work are as follows:

- We propose a novel approach that separates cross-scene correspondence extraction from policy learning, empowered by a correspondence-aware policy conditioned on the extracted explicit correspondences, thereby enhancing flexibility and scalability.

- We design a novel graph neural network that is biased to leverage direct relationships from explicit correspondences, improving the model's generalization across tasks and objects.

- We introduce an efficient online adaptive matching algorithm that dynamically provides reliable correspondences during execution, enabling precise policy adaptation in real world.

- We conduct extensive evaluations of our method in both simulation and real-world settings. The results demonstrate that the proposed approach achieves strong few-shot performance, while also exhibiting robust cross-instance and even cross-category generalization.

## 2 RELATED WORKS

### 2.1 MULTI-TASK IMITATION LEARNING

A growing body of work focuses on developing generalist robot policies (Kim et al., 2024; Team et al., 2024; Jang et al., 2022; Black et al.; Liu et al., 2024; Cheang et al., 2024). These models, typically conditioned on visual observations and language instructions, are trained on large-scale robotic datasets in real world (O'Neill et al., 2024; Khazatsky et al., 2024; Bu et al., 2025) or in simulation (James et al., 2020; Mu et al., 2024) and achieve strong average performance on the tasks seen during training. With appropriate visual inputs and language prompts, they can generalize zero-shot to related tasks and scenes under varying visual conditions. However, their performance often degrades on entirely novel tasks, additional fine-tuning is usually required to attain acceptable results. Our method bypasses the task generalization obstacle by directly conditioning the policy on demonstration, which showcases the fine-grained intention of the task more effectively.

### 2.2 IN-CONTEXT IMITATION LEARNING

In-context learning (ICL) has emerged as a powerful paradigm, enabling models to adapt to novel tasks of language (Brown et al., 2020), vision (Zhou et al., 2024) and sequential decision making (Raparthy et al., 2023) from only a handful of demonstrations without any retraining. In robotics, explicit ICL-based approaches often begin by establishing correspondences between given demonstrations and a new scene, followed by transforming the trajectories using handcrafted heuristics (Zhang & Boularias, 2024; Tang et al., 2025; Zhu et al., 2024; Heppert et al., 2024). While effective in controlled settings, they typically demand considerable human effort and their adaptability to truly novel scenarios remains constrained. The in-context learning ablity from Large Language Models (LLMs) has also be adapt for robot (Di Palo & Johns, 2024; Yin et al., 2024). However, these methods usually run in open-loop and face difficulty to efficiently predict accurate motions

Recent studies have pursued training in-context policy models directly, leveraging transformers (Jain et al., 2024; Fu et al., 2024) or graph neural networks (Vosylius & Johns, 2024) to minimize human involvement. In contrast, MatchingPolicy adopts a direct and modular strategy: it explicitly extracts correspondence information from Vision Foundation Models (VFMs), freeing the policy to focus solely on adapting robot actions to the new environment. This decoupled design not only reduces manual intervention but also enhances generalization to novel objects.

Several recent works (Sridhar et al., 2025; 2024) employ a similar modular design, decoupling relationship extraction between demonstrations and the current scene from action prediction using retrieval-augmented generation. In contrast, our method extracts dense point-wise correspondences,

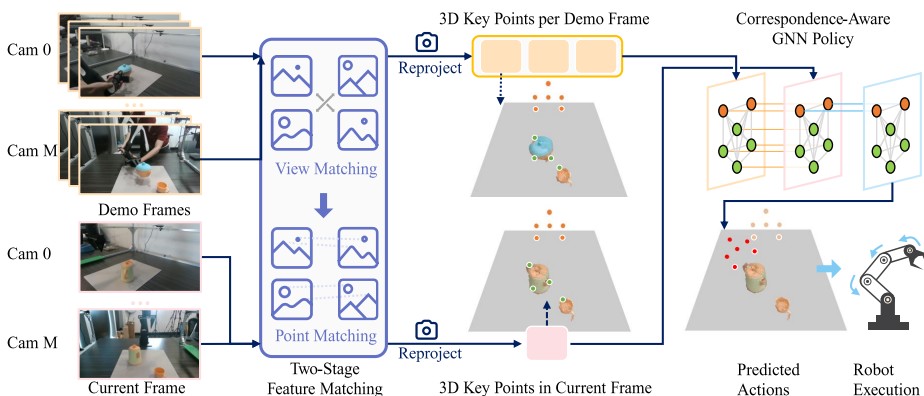

Figure 1: The overview of MatchingPolicy. Correspondence points are extracted using a two-stage feature matching algorithm and passed into a graph-based diffusion policy model, which predicts the 6D motions of the gripper. These motions are subsequently converted into robot joint commands.

which not only provides a similarity measure but also delivers fine-grained spatial cues that facilitate 3D robot action prediction.

### 2.3 KEYPOINT-BASED POLICY LEARNING

Keypoint-based representations are widely used in robotics (Puang et al., 2020; Gao et al., 2023) for their sparsity and interpretability. They also serve as high-level abstract action representations, either predicted by visual large language models (VLLMs) (Huang et al., 2024b;a) or generated by learned policies (Wen et al., 2023; Bharadhwaj et al., 2024; Xu et al., 2024), and can be mapped to low-level robotic control, enabling VLLM-driven manipulation and cross-embodiment policy transfer. Recently, several works have integrated keypoint-based representation into policy training as observation (Haldar & Pinto, 2025; Wang et al., 2024; Levy et al., 2024; Wang et al., 2025; Fang et al., 2025), demonstrating improved generalization to novel scenes and objects and improving the data efficiency. These methods typically rely on pre-defined keypoints, either selected manually or determined via unsupervised clustering, which can limit the policy's adaptability to novel scenarios. Our work differs in two key aspects: 1) Task Scope: prior works focus on single-task imitation learning, whereas MatchingPolicy targets in-context multi-task learning; 2) Key Point Adaptation: instead of static pre-defined key points, our method dynamically generates key points based on matching confidence and visibility. This eliminates human effort while ensuring better feature alignment.

## 3 METHOD

**Problem Formulation** We address robotic manipulation in a few-shot, in-context learning setting. Given a small set of $N$ demonstrations $\{D_j\}_{j=1}^N$, where $N \in \{1, 2\}$, the goal is to leverage these demonstrations to perform the same task depicted in the demos, but in a novel scene.

The policy observation of the current scene is represented by a state $s_c = (P_c, g_c)$. Here, $P_c$ denotes the 3D point cloud, obtained from RGB-D images and transformed into the gripper's local coordinate frame via the gripper pose $T_c \in \text{SE}(3)$, while $g_c \in \{0, 1\}$ represents a binary gripper state. Similarly, a demonstration $D_j$ is a time-indexed sequence of $L$ states, $D_j = (s_{j,t})_{t=1}^L$, where each $s_{j,t} = (P_{j,t}, g_{j,t})$ follows the same definition as $s_c$. Given the current state $s_c$ and the demonstration set $\{D_j\}_{j=1}^N$, the policy $\pi$ predicts a horizon of $K$ future actions $(a_k)_{k=1}^K$, where $a_k = (\Delta T_k, g_k)$ comprises a relative gripper transformation $\Delta T_k$ and a target gripper state $g_k$. Formally, the objective is to learn a policy that models the conditional distribution $\pi = p(a_{1:K} \mid s_c, (D_j)_{j=1}^N)$. We use the term "frame" to refer uniformly to the current state, a demonstration state, or a predicted action, drawing a direct analogy to frames in a video.

**Method Overview** The pipeline of our method is shown in the Fig. 1. Our key insight is to decouple correspondence extraction from action prediction. Specifically, we first explicitly establish

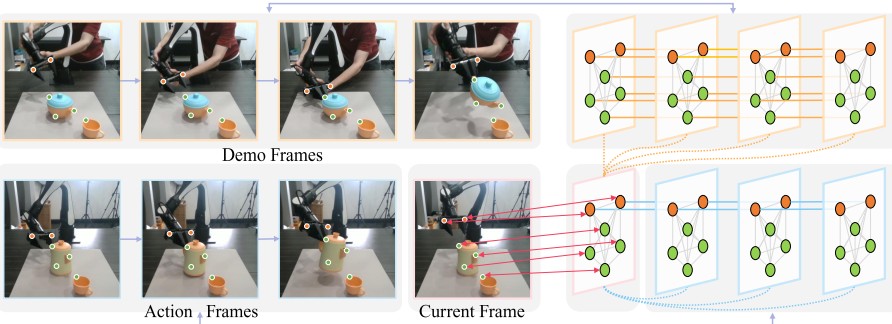

Figure 2: Our Correspondence-Aware Policy Model. Left: The randomly selected correspondence key points on 2D images of demonstration frames, current frame, and action frames. Right: The structure of our graph-based policy model. The red lines with two-way arrow indicate the one-to-one mapping between the graph nodes in the network and the key points in the current frame. The lines connecting nodes or frames indicate message passing between them.

semantic correspondences by identifying a set of key points in the current frame and locating their counterparts across all demonstration frames. These matched point sets, which encapsulate robust spatio-temporal information, serve as the structured context for our in-context imitation policy. The policy then leverages this context to predict precise end-effector motions, which are subsequently translated into robot actions. We introduce a correspondence-aware policy that efficiently leverages provided semantic correspondences to generate robot actions (Section 3.1). The policy is trained on synthetic demonstrations with ground-truth correspondences (Section 3.2). For deployment in real-world settings, we introduce a two-stage feature matching algorithm that utilizes off-the-shelf VFMs for real-time extraction of reliable correspondences (Section 3.3).

**Point Selection for Correspondence**   We select points from the current observation's point cloud based on two criteria: first, each point must have a semantically matched correspondence across most demonstration frames; second, the final set must be both representative of the scene and sufficiently random for robustness.

To this end, we employ a two-step selection process. First, for every point in the current observation, Points that meet the first criterion are retained as candidates. Second, we apply Farthest Point Sampling (FPS) to these candidates to select a final, spatially distributed set of points. Their corresponding points in the demonstration frames form the representative set for the demonstrations. More specifically, we compute these correspondences using template matching during training (Section 3.2) or a two-stage matching process during execution (Section 3.3)

## 3.1 Correspondence-Aware Diffusion Policy via Graph Neural Network

Instead of directly using the full scene point cloud as input, our policy model selects a set of key points from the point cloud, which are encoded with geometric features. Additionally, key points from the gripper are included to represent its pose. The model assumes that the key points selected from $P_{j,t}$ in the demonstration frames correspond to those from $P_c$ in the current frame. However, these correspondences may be imperfect due to occlusions or errors introduced by the VFMs.

To fully leverage the assumed correspondences between the demonstrations and the current scene—and to predict actions consistent with the demonstrated intent, we introduce a heterogeneous graph representation, which jointly encodes the demonstrations, current observations, and predicted actions, modeling their complex interactions, ensuring critical information flows to where it's needed for precise action prediction.

The model structure is shown in the right of Fig. 2. We group the nodes according to demonstration frames $\{D_j\}_{j=1}^N$, the current frame $s_c$, and action frames $(a_k)_{k=1}^K$ which represents the future states of current scene. Each frame includes $M_g = 6$ gripper nodes and $M_s = 16$ scene nodes. The bidirectional red arrows represent a one-to-one mapping between the graph nodes and the key points

in the current frame. In both the demonstration and current frames, gripper nodes store the 3D positions of key points and gripper state, while scene nodes store only the 3D positions. In action frames, gripper nodes store the displacement of gripper nodes relative to the current frame, and scene nodes replicate those from the current frame. Once the displacements are predicted, we convert them into the gripper's relative pose transformation $\Delta T_k$ using Singular Value Decomposition (SVD) (Arun et al., 1987). Lines without arrow represent edges for propagating information in graph neural network, with the key distinction being that a solid line between nodes denotes a single relational edge between two corresponding nodes of two frames, while a dotted line between frames represents a group of point-to-point edges for each pair of corresponding nodes.

**Spatial Structure**   To extract the spatial structure in the same frame, we build a fully-connect graph among nodes of the gripper and the scene, in demonstration frames, current observation, and action frames, shown as gray lines within frames in Fig. 2. We assign relative positions between nodes in Cartesian space as the edge attributes. Positional encoding (Vaswani et al., 2017) on the relative positions is used to precisely capture high-frequency details.

**Direct Corresponding**   Since the matched scene key points and selected gripper points have direct correspondences across frames, we connect the gripper and scene nodes in the current frame to their corresponding nodes in the demonstration frames. Additionally, we establish connections between corresponding nodes in adjacent frames. These connections, illustrated as orange lines in Fig. 2, while the orange dotted lines between frames indicate that the point-to-point connections are omitted for clarity. This connections facilitate the efficient propagation of spatial correspondences from the demonstrations to the current observation, thereby simplifying the action prediction process.

**Action Prediction**   To propagate essential information for action prediction, we connect the gripper nodes in each action frame to their corresponding gripper nodes in the current observation frame (shown as blue dotted lines indicating point-to-point between gripper nodes), as well as to those in the preceding action frames (shown as blue solid lines).

**Local Geometric Encoder**   We employ a geometric encoder to propagate geometric information from unselected points to the key points. Our geometric encoding process consists of two stages: point association and feature fusion. First, we establish local neighborhoods by assigning each unselected point to its geometrically closest key point. These neighborhood associations then serve as input to a local PointNet encoder (Qi et al., 2017), which was pre-trained using a geometric reconstruction objective (Mescheder et al., 2019). The encoder processes these local point groupings to extract and consolidate geometric features.

**Graph-based Diffusion Model Training**   We adopt the training strategies from (Vosylius & Johns, 2024) and train the model via a diffusion process. Given the ground truth gripper's nodes' position and state in the action frames, noise $\epsilon$ is iteratively added to the positions and states, transforming them into a normal distribution. The graph model is trained to predict the noise $\epsilon_\theta = [\nabla p, \nabla a]$ for each gripper node in action frames, then applied to the gripper nodes to reconstruct the original positions and states. Here $\theta$ represents the parameters of the graph neural network, and $\nabla p$ can be interpreted as the displacement from the noisy positions to the ground-truth positions. To improve the prediction of fine-grained displacements caused by rotation and prevent translation from dominating small rotations, the displacement prediction are decomposed into two components: $\nabla p = [\nabla p_t, \nabla p_R]$, where $\nabla p_t$ is the displacement induced by the gripper's translation, and $\nabla p_R$ represents the displacement due to rotation. Thus, we learn $\epsilon_\theta = [\nabla p_t, \nabla p_R, \nabla a] \in \mathbf{R^7}$ for each gripper node in action frames and optimize the variational lower bound of the data likelihood, which is equivalent to minimizing the mean squared error $MSE(\epsilon_\theta - \epsilon)$ according to Ho et al. (2020).

### 3.2 PSEUDO DEMONSTRATION SYNTHESIS WITH GROUND-TRUTH CORRESPONDENCE

Our policy leverages in-context imitation learning (Fu et al., 2024), which infers task intent dynamically from context. A key advantage of this paradigm is that the model's weights are not specific to any given task. This inherent task-agnosticism allows us to generate datasets with both executional variability and semantic consistency in simulation entirely.

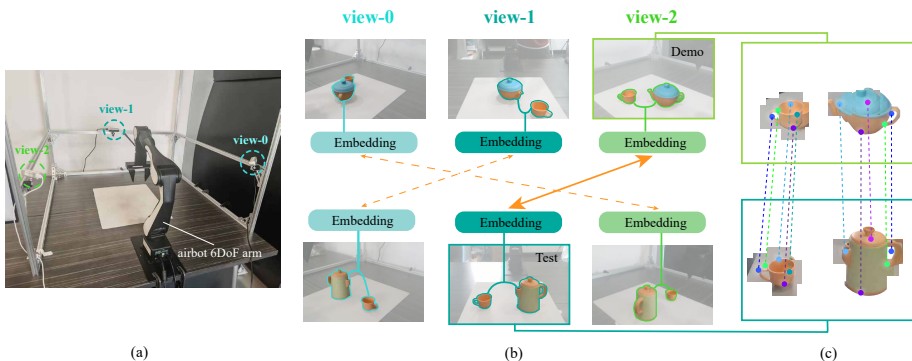

view-0      view-1      view-2

(a)                         (b)                         (c)

Figure 3: Two-stage Feature Matching Algorithm. (a) Camera setup (b) View matching based on the average features of segmented objects. (c) Point matching on objects for matched view pairs.

**Training Data Generation** We generate these semantically consistent yet diverse datasets by defining tasks that span various motion primitives (e.g. pick-and-place, grasping, push-pull, random path following.). For each task, we randomly sample objects from ShapeNet (Chang et al., 2015), define object-centric waypoints for the gripper, and simulate interaction physics by rigidly attaching the object to the gripper upon closure. By systematically randomizing parameters (e.g. initial gripper and object pose), we create demonstration sets that pair high instance-level variability with a consistent task-level goal. Please refer to Appendix A for more details.

**Key Point Candidates and Selection via Ground Truth Correspondence** Knowing the exact object poses and models in simulation allows us to unambiguously find point correspondences between the current frame and the demonstration frames. We first sample template points $P^{gt}$ on the mesh of the object as anchor for matching. For observed point cloud in each frame, we can find its closest points $P^{obs}$ to the template points. if $(\|p^{obs} - p^{gt}\| < 0.02\,\text{cm})$, we think the $p^{gt}$ is **visible** in that frame. The $p^{obs}$ correspond to the same template point $p^{gt}$ are considered **matched**.

We construct correspondence point candidates from template points using two criteria: 1) its **matched point** in current frame is **visible**; 2) More than half of the **matched point** in demonstration frames are **visible**. In each policy training step, template points are randomly sampled from the candidate set using farthest point sampling (FPS), and their correspondences across demonstration frames are used as key points. When a matched point is not **visible** in a demonstration frame, we resample a point from the observed cloud using FPS. This intentional introduction of incorrect correspondences serves as data augmentation, enhancing the policy model's robustness.

### 3.3 REAL-WORLD CORRESPONDENCE INTEGRATION VIA FOUNDATION MODELS

An advantage of our method is its plug-and-play capability to incorporate off-the-shelf vision foundation models during test time, thereby enhancing policy generalization. During deployment in a novel scene, our method begins by densely sampling candidate points in the initial frame and then leverages a lightweight matching algorithm alongside VFMs (Heinrich et al., 2025; Oquab et al., 2023) to establish their correspondences in the demonstration's initial frames. To capture temporal correspondences, these points are propagated across subsequent demonstration frames and tracked online during policy execution using a visual point tracking model (Karaev et al., 2024). We use the same correspondence key point selection criteria as described in Section 3.2,

**Inter-scene Correspondence via Vision Foundation Model** For robust 3D semantic matching between demonstration and test scenes, we introduce a two-stage algorithm that leverages powerful 2D VFM priors. It first establishes the most similar images from different views and then finds detailed 2D point correspondences, which are subsequently projected into 3D space.

As shown in Fig. 3(a), our method is designed for a multi-camera setting—such as the three-camera, 360° setup in our experiments—to explicitly resolve pose ambiguity. The core challenge is that an

object in a test scene may have different canonical orientations from its counterpart in the demonstration. Our **Adaptive View Matching** algorithm tackles this by identifying the optimal alignment between the demonstration and testing views. It evaluates three discrete rotational hypotheses (no rotation, $120°$ left/right rotation) by establishing a unique view-to-view mapping for each. For example, in Fig. 3(b), a $120°$ right rotation of the test object implies that the demonstration *view-1* should be matched with the test *view-0*, *view-2* with *view-1* and so on. For each hypothesis, the algorithm computes the average cosine feature similarity across all corresponding view pairs. The hypothesis yielding the highest similarity score is selected as the optimal alignment. Once this alignment is determined, we perform 2D feature point matching (Heinrich et al., 2025) between the corresponding image pairs and project the resulting correspondences into 3D space using the camera's intrinsic and extrinsic parameters as shown in Fig. 3(c).

**Intra-scene Correspondence via Visual Tracking Model** Following prior methods (Haldar & Pinto, 2025; Wang et al., 2025), we utilize an off-the-shelf point tracking model (Karaev et al., 2024) to establish temporal correspondences. Unlike previous methods that only track sparse, expert-annotated, or clustered keypoints, our approach tracks dense points on the objects. Similarly, for each point candidate, we also maintain a **visibility** state: a point is considered **visible** if (1) it is present in the frame, as indicated by the tracking model, and (2) its depth value is valid, accounting for missing pixels from depth sensors. Tracking is performed independently across different camera views, and the tracked points are subsequently projected into the 3D world coordinate system (aligned with the robot base frame). We track $N_{track} = 512$ points per camera view.

### 3.4 IMPLEMENT DETAILS

**Training/Test Details** We trained our model using the AdamW optimizer with a learning rate of $1 \times 10^{-5}$. It took seven days on four NVIDIA A6000 GPUs to traing a model for 1M steps. All evaluation was performed on a PC with a RTX 4080 GPU.

## 4 EXPERIMENTAL RESULTS

In this section, we address the following questions: 1)Does `MatchingPolicy` outperform baseline methods overall as it can better utilize explicit correspondences? 2) Can `MatchingPolicy` leverage an off-the-shelf large vision model to facilitate generalization in real-world scenarios?

### 4.1 EXPERIMENT SETUP

**Benchmarks and Baselines** We first evaluate our method on 36 RLBench (James et al., 2020) simulation tasks, where identical objects are arranged in diverse initial configurations. We divide them into three categories: 1) 12 Fine-Tuning Tasks: these tasks are used to fine-tune models to reduce domain gap. 2) 12 Similar Tasks: these tasks share similar objects' geometries to the Fine-Tuning Tasks (e.g., close box vs. open box). 3) 12 Unseen Tasks: the objects are entirely unseen (e.g. open door) or the tasks are different but share similar geometries with previously seen ones which may confuse the policy (e.g., opening the lid of a grill vs. placing meat on the grill). To provide a comprehensive evaluation, we compare our method against three representative approaches: InstantPolicy (Vosylius & Johns, 2024), a geometric in-context learning baseline using Graph Neural Networks; KAT (Di Palo & Johns, 2024), a keypoint-based method leveraging LLM reasoning for action generation; and RDT-1B (Liu et al., 2024), a state-of-the-art Vision-Language-Action (VLA) model evaluated in both zero-shot and fine-tuned settings.

**Model Fine-tuning** Since MatchingPolicy is initially trained solely on pseudo-demonstrations, we address the domain gap by fine-tuning the model with additional demonstrations collected either in RLBench or the real world. The fine-tuning process employs a balanced 50/50 mixture of pseudo-demonstrations and newly collected RLBench simulated demonstrations.

### 4.2 SIMULATION EXPERIMENTS

We compare our method with the baseline approaches on the RLBench simulation tasks, as summarized in Table 1. Ground-truth correspondences in RLBench are obtained following the procedure

Table 1: Success rates of our method (OURS) versus the baseline (IP) on 36 RLBench tasks. Fine-tuning tasks are tasks used to retrain the model, similar tasks are structurally related, and unseen tasks were provided only as context. For the first two categories, scores are presented as before/after fine-tuning. For unseen tasks, the scores reflect performance after the model has been fine-tuned on the fine-tuning task set.

| Fine-tuning Tasks | | | Similar Tasks | | | Unseen Tasks | | |
|---|---|---|---|---|---|---|---|---|
| **Task** | IP | OURS | **Task** | IP | OURS | **Task** | IP[1] | OURS |
| Open Box | 0.94/0.99 | 1.00/1.00 | Slide Buzzer | 0.35/0.94 | 0.71/0.74 | Close Drawer | 0.00 | 0.90 |
| Close Jar | 0.58/0.93 | 0.78/0.92 | Plate Out | 0.81/0.97 | 0.96/0.99 | Close Grill | 0.00 | 0.97 |
| Toilet Seat Down | 0.85/0.93 | 1.00/1.00 | Close Laptop | 0.91/0.95 | 1.00/1.00 | Open Grill | 0.00 | 0.99 |
| Close Microwave | 1.00/1.00 | 0.97/1.00 | Close Box | 0.77/0.99 | 1.00/1.00 | Pick Up Cup | 0.00 | 0.94 |
| Phone on Base | 0.98/1.00 | 1.00/1.00 | Open Jar | 0.52/0.78 | 0.89/0.91 | Open Door | 0.01 | 0.98 |
| Lift Lid | 1.00/1.00 | 1.00/1.00 | Toilet Seat Up | 0.94/1.00 | 1.00/1.00 | Turn Tap | 0.00 | 1.00 |
| Take Umbrella Out | 0.88/0.91 | 0.92/1.00 | Meat off Grill | 0.77/0.90 | 0.82/0.89 | Frame Off | 0.00 | 0.33 |
| Slide Block | 0.75/1.00 | 0.98/1.00 | Open Microwave | 0.23/0.56 | 0.43/0.54 | Sweep Dust | 0.00 | 0.41 |
| Push Button | 0.60/1.00 | 0.89/1.00 | Paper Roll Off | 0.70/0.95 | 0.94/0.96 | Open Wine | 0.01 | 0.71 |
| Basketball in Hoop | 0.66/0.97 | 0.87/0.97 | Put Rubbish in Bin | 0.97/0.99 | 1.00/1.00 | Put Money | 0.00 | 0.50 |
| Meat on Grill | 0.78/1.00 | 0.83/1.00 | Put Umbrella | 0.31/0.37 | 0.33/0.40 | Put roll | 0.00 | 0.52 |
| Flip Switch | 0.40/0.94 | 0.90/0.99 | Lamp On | 0.42/0.41 | 0.81/0.85 | Unplug Charger | 0.00 | 0.46 |
| **Average** | 0.71/0.97 | 0.92/**0.99** | **Average** | 0.60/0.82 | 0.83/**0.86** | **Average** | 0.00 | **0.73** |

[1] We test Instant Policy using the official checkpoint.

Table 2: Comparison between MatchingPolicy against KAT (a LLM-based In-context learning method) and RDT-1B (a representative VLA method) under six challenging RLBench tasks. Our method achieves superior performance with only 2 demonstrations.

| **RLBench Tasks** | **KAT (LLM-based ICL)** | | **RDT-1B (VLA)** | | **MatchingPolicy (OURS)** |
|---|---|---|---|---|---|
| | Aligned Setting (2 demos) | Original Paper (10 demos) | Zero-shot (0 demo) | Task-Specific FT (20 demos) | (2 demos) |
| Plate Out | 0.12 | 0.36 | 0.08 | 0.67 | 0.96 |
| Slide Buzzer | 0.04 | 0.19 | 0.00 | 0.47 | 0.71 |
| Toilet Seat Up | 0.25 | 0.38 | 0.12 | 0.69 | 1.00 |
| Meat off grill | 0.33 | 0.54 | 0.05 | 0.73 | 0.77 |
| Pick up Cup | 0.28 | 0.47 | 0.09 | 0.81 | 0.94 |
| Unplug charger | 0.00 | 0.12 | 0.00 | 0.32 | 0.46 |

described in Section 3.2. Our method achieves performance comparable to or better than the baseline across the tasks. A key advantage of our approach is its superior generalization capability. Without requiring any fine-tuning, our policy achieves high success rates, even surpassing the baseline's performance after it has been fine-tuned on specific tasks (e.g., "*Take Umbrella Out*," "*Paper Roll Off*"). This generalization gap is most evident on the challenging Unseen Tasks, where the baseline fails completely, even when some objects were present in the Fine-tuning Tasks. We hypothesize that this is because the baseline method (IP) relies heavily on object geometry to infer task intent, which inherently limits its task generalization. In contrast, by conditioning on demonstration-scene correspondences, MatchingPolicy is less dependent on object geometry and consequently achieves strong performance on unseen tasks.

We further benchmark MatchingPolicy against representative LLM-based In-context Learning method ( Di Palo & Johns (2024)) and VLA method ( Liu et al. (2024)) on six challenging tasks (Table 2). MatchingPolicy demonstrates superior data efficiency: with only 2 demonstrations, it consistently outperforms KAT even when the latter utilizes 10 demonstrations. This highlights the advantage of explicit correspondence over pure LLM reasoning for fine-grained manipulation. Furthermore, while the VLA model struggles with zero-shot transfer and requires 20 demonstrations to recover performance, MatchingPolicy (2 demos) still yields higher success rates. This confirms our framework's ability to achieve superior performance without the computational cost of VLA fine-tuning.

Table 3: Results on real-world tasks, evaluated across different levels of generalization.

| Level | Method | Pour Water | Put Lid | Open Box | Cut Egg | Average |
|-------|--------|-----------|---------|----------|---------|---------|
| Layout-easy | InstantPolicy | 0/10 | 9/10 | 0/10 | 0/10 | 22.5% |
| | MatchingPolicy | 7/10 | 8/10 | 9/10 | 7/10 | 77.5% |
| Layout-hard | InstantPolicy | 0/10 | 0/10 | 0/10 | 0/10 | 0.00% |
| | MatchingPolicy | 6/10 | 6/10 | 8/10 | 6/10 | 65.0% |
| Shape | InstantPolicy | 0/10 | 4/10 | 0/10 | 0/10 | 10.0% |
| | MatchingPolicy | 8/10 | 6/10 | 8/10 | 7/10 | 72.5% |

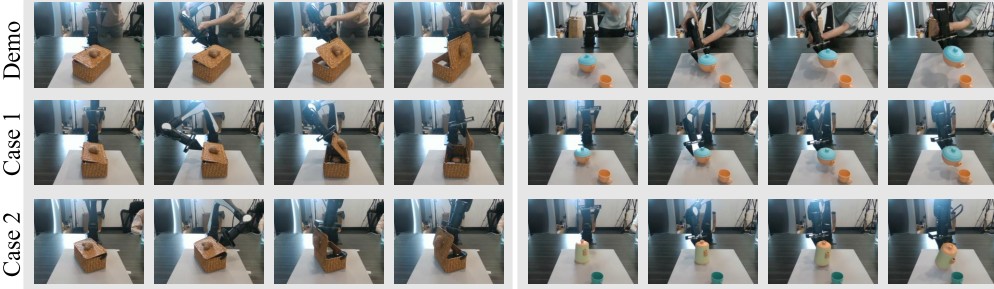

Figure 4: Generalization capabilities of MatchingPolicy from a single demonstration. The policy adapts to novel scene layouts (Left) and unseen object instances (Right).

## 4.3 REAL-WORLD EXPERIMENTS

In our real-world experiments, we evaluate our method's ability to master everyday manipulation skills and generalize to novel objects with variations in appearance and geometry. The experiments are carried out using an AirBot Pro 6-DoF robotic arm [1]. The scene is observed through three Intel RealSense D435i RGB-D cameras. To validate the models' potential for sim-to-real generalization across diverse scenarios, we deploy them directly without real-world fine-tuning.

**Real-World Tasks and Variants** To rigorously test generalization, we define three difficulty levels of generalization: **Layout-Easy**: Objects are the same in test scene as in the demonstrations, with minor pose variations. **Layout-Hard**: Similar to Layout-Easy, but with significant pose variations compared to the demonstrations (e.g., objects rotated to face completely opposite directions); **Shape**: Objects in test scene differ in shape and appearance from those in the demonstrations. We provide more details in Appendix B. For each task and generalization level, we conduct ten trials and report the success rates in Table 3. We observe MatchingPolicy successfully completes all tested tasks, achieving a higher success rate and outperforming the baseline across all generalization levels.

**Generalization to novel layout** We observe that the success rate of InstantPolicy on the *Put Lid* task in the Layout-Hard setting drops to 0%. Relying on an coarse geometric representation and biased by the demonstration trajectory, it fails by grasping the pot instead of the lid. In contrast, our approach leverages VFMs to extract precise visual and semantic correspondences, enabling MatchingPolicy to identify the lid and complete the task. In more challenge task *Open Box*, MatchingPolicy can generate successful trajectories that can approach objects from directions totally different from that in the demonstration (Fig. 4). This demonstrates that our method follows the motion of matched keypoints instead of merely replaying the demonstrated trajectory.

**Generalization to novel objects** In the **Shape** generalization setting, MatchingPolicy again significantly outperforms the baseline, achieving a 72.5% average success rate. As illustrated on the right of Fig. 4, for example, MatchingPolicy successfully completes the *Pour Water* task even when the teapot differs substantially in shape. Remarkably, the policy also demonstrates cross-category

---
[1]https://airbots.online/

generalization on the **Put Lid** task, correctly placing a lid on a teapot after seeing only a demonstration of placing a lid onto a pot (see Appendix B). These results confirm that our method relies on learned correspondences and is largely agnostic to object geometry.

## 4.4 ABLATION STUDIES

We conduct comprehensive ablation studies by comparing several variants of our approach. We evaluate our method against variants using a naive matching approach in real-world scenarios. Additionally, we explore test-time planning strategies to further enhance our policy's performance.

**Two-Stage Matching versus Naive 2D-to-3D Matching** We compare our two-stage matching approach against a 3D naive matching baseline. The baseline, based on Wang et al. (2024), projects 2D features into the world frame and fuses them by their 3D coordinates before establishing correspondences. As shown in Table 5, our method achieves superior policy performance by generating more accurate semantic correspondences. We ob-

| Matching | Planning | Layout-Easy | Layout-Hard | Shape |
|----------|----------|-------------|-------------|-------|
| Naive | Overlap | 37.5% | 5.0% | 10.0% |
| 2-Stage | Whole | 54.0% | 47.5% | 57.5% |
| 2-Stage | Half | 60.0% | 45.0% | 45.0% |
| 2-Stage | Overlap | 77.5% | 65.0% | 72.5% |

Figure 5: Average success rates comparing our full method (bottom row) to degraded variants.

serve that the performance of the naive baseline drops significantly in the **Shape** and **Layout-Hard** settings, as the naive matching generate too much wrong correspondences which mislead the policy and lead to task failure.

**Test-time Planning Methods** we develop an additional planning strategy adapted from Janner et al. (2022) to improve test-time performance: After the policy predicts $K$ actions, the robot executes the first $K/2$ actions. The remaining $K/2$ actions seed the next prediction by replacing the first $K/2$ action of the next action chunk, resulting in an inference process where the first half of an action chunk is denoised for only the final diffusion step, while the second half undergoes the full diffusion process. We term "Overlapping chunk" and used it in other real-world experiments. We compare it with against several alternatives: 1) Whole chunk ("Whole"): the robot executes all $K$ predicted actions. 2) Half chunk ("Half"): the robot executes the first $K/2$ steps and then re-predicts. The results in Fig. 5 demonstrate that "overlap" significantly enhances a policy's performance. In addition, the planning method improves the smoothness of the resulting trajectories.

## 5 CONCLUSIONS AND LIMITATIONS

**Conclusions** In this work, we introduced MatchingPolicy, a novel correspondence-aware policy that transforms robot actions from demonstrations to new scenes using dense correspondence cues. By leveraging off-the-shelf vision foundation models with a novel two-stage matching algorithm, our approach enables one-shot imitation on tasks with unseen objects and different layout in the real world. Our work demonstrates that explicitly modeling the visual and semantic relationships between the demonstration and the current scene is key to effective in-context imitation learning. This principle of learning from analogy, rather than from vast datasets of robot trials, paves the way for building more generalizable and data-efficient robotic systems.

**Limitations** Our method also has several limitations. First, we focus primarily on short-horizon manipulation tasks; extending our approach to long-horizon tasks would be a valuable direction for future work. Second, our method currently only consider semi-dynamic scenes and operates at a relatively slow speed, making it difficult to handle highly dynamic tasks such as fling (Ha & Song, 2022). Third, our view selection process is somehow dependent on our current platform, requiring adaptation to be applied effectively on other platforms. We plan to address these limitations and improve our method in future work.

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

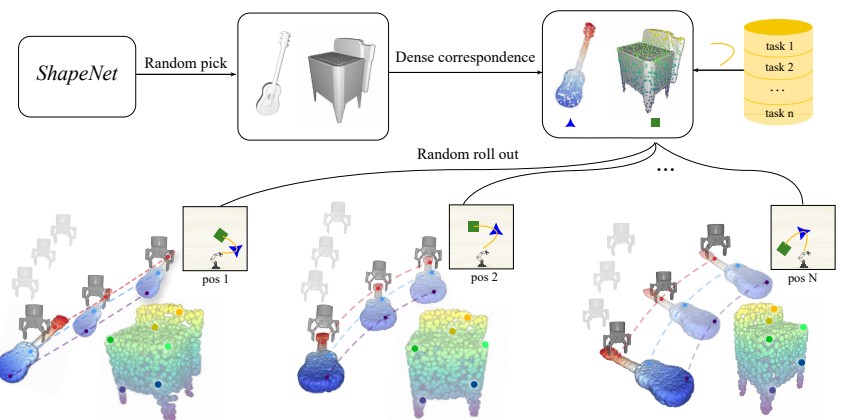

Figure 6: Pesudo Data Generation Pipeline

APPENDIX

# A  TRAINING DATA DETAILS

**Training Data Generation**  To generate a large-scale dataset of semantically consistent pseudo-demonstrations without relying on real, task-specific data, we developed the following data generation pipeline, as illustrated in Fig. 6.

1. **Task and Object Initialization:** For each data sample, we first select a manipulation task from a predefined pool containing diverse motion primitives (e.g., pick-and-place, push-pull, open-close, and structured random movements). We then randomly sample two or three object models from the ShapeNet dataset and assign them randomized scales.

2. **Rule-Based Waypoint Definition:** We define a set of object-centric key waypoints for the gripper trajectory based on the task goal. This rule-based approach ensures semantic consistency across different demonstrations. For example, in a **"pick-and-place"** task involving objects A and B:

    a. The *picking waypoint* is defined as a randomly sampled point on the surface mesh of the source object (A), with the gripper approaching along the surface normal.

    b. The *placing waypoint* is defined as a randomly sampled point on or near the surface of the target object (B).

3. **Gripper State and Attachment:** Gripper actions (open/close) are scheduled at the appropriate waypoints. Once the gripper closes at the picking waypoint, the picked object (A) is rigidly attached to the gripper's coordinate frame for subsequent movement.

4. **Full Trajectory Interpolation:** We use interpolation algorithms (e.g., cubic, linear, spherical) to generate a smooth, continuous gripper trajectory that passes through the defined waypoints. This interpolated path, along with the corresponding gripper states, forms a single pseudo-demonstration.

5. **Generating Intra-Task Variations:** To create a diverse dataset for a single conceptual task, we generate multiple demonstrations by randomizing the initial poses (position and orientation) of the objects. Critically, the object-centric picking and placing locations remain consistent *relative to the objects' own frames*. For example, the policy always learns to pick object A from a point *on its surface* and place it *on object B*, regardless of where A and B are in the workspace.

By combining randomized object instances and initial poses with semantically consistent, object-centric waypoints, this pipeline produces a rich dataset that pairs high instance-level variability with

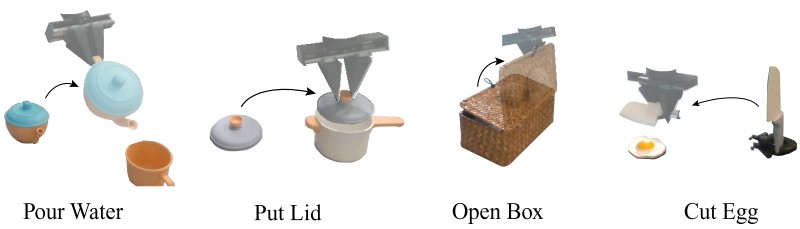

Figure 7: The real-world tasks for evaluation.

a consistent task-level goal, which is crucial for training a generalizable policy. Specifically, our dataset comprises 52k pseudo-task groups (where each group contains 3 rollouts with randomized instances and poses), totaling approximately 156k trajectories. This includes 10k groups for pick-and-place, 10k for open-and-close, 6k for grasping, 6k for push-pull, and 20k for random path following.

**Key Point Candidates and Selection via Ground Truth Correspondence** A key advantage of our synthetic environment is access to ground-truth object poses and meshes for identical objects across trajectories. This enables the unambiguous extraction of point correspondences between the current frame and demonstration frames. For clarity, we denote the set of observed point clouds as $(P_i^{obs})_{i=0}^{NL+1}$, where $i = 0$ corresponds to the current frame.

Given the observed point cloud, along with the poses of the objects, we begin by uniformly sampling a total of $W = 2048$ template points from the objects' meshes. These points are transformed into the world coordinate frame, resulting in the transformed template point clouds $(P_i^{gt})_{i=0}^{NL+1}$. Since the template points are identical in the object local coordinate and can serve as anchors for matching correspondence. For each transformed template point cloud $p_i^{gt} \in P_i^{gt}$, We search for its nearest point $p_i^{near}$ in the observed point clouds $P_i^{obs}$. If the distance between $p_i^{near}$ and $p_i^{gt}$ is less than a threshold $r = 0.02cm$, we consider the point pair $(p_i^{gt}, p_i^{near})$ to be **matched** and $p_i^{near}$ is **visible**. Furthermore, as a template point $p^{gt}$ may have nearest match points $p_i^{near}$ in different frame $i$, we denote the matching set of the $p^{gt}$ as $\{p_i^{near}\}_{i=0}^{NL+1}$. We compute the matching set for each template point. Given $NL + 1$ frames for matching, each matching set has up to $NL + 1$ nearest observed points. We also store a flag array $(f_i^{gt})_{i=0}^{NL+1}$ for each template point $p_i^{gt}$. The flag indicate whether the $p_i^{near}$ is **visible**.

As our model explicitly rely on $M_g$ points as the input for each frames. We first retain only those template points $p^{gt}$ that satisfy both of the following conditions: 1) The $f_i^{gt} = True$ for $i == 0$; 2) More than half of the nearest observed points $p_i^{near}$ across all $NL+1$ frames are **visible**. Let $\{p^{cand}\}$ be the set of surviving template points. We then apply farthest point sampling (FPS) to $\{p^{cand}\}$ to select $M_s$ representative key points. For each selected point $p^{cand}$, their nearest observed points $\{p_i^{near}\}_{i=0}^{NL+1}$ are regarded to share correspondence and used as the input of the policy model. For the $p_i^{near}$ in frame $i$ where $f_i^{gt} = False$, we instead resample a point from the observed point cloud $P_i^{obs}$ using FPS. These wrong correspondence serve as a data augmentation for the policy train, and increasing the robustness of the policy model.

## B  REAL-WORLD EXPERIMENT DETAILS AND RESULTS

**Three-level Generalization** We assess our models on a set of challenging everyday manipulation tasks, as shown in Fig. 7. The variants of the objects are show in Fig. 8.

1. **Layout-Easy:** We randomize the scene by rotating all objects between $-30°$ and $30°$ and applying a random translational offset of 3–5 cm on the table plane.

2. **Layout-Hard:** We apply more aggressive randomization, rotating objects between $90°$ and $150°$ with the same 3–5 cm translational offset.

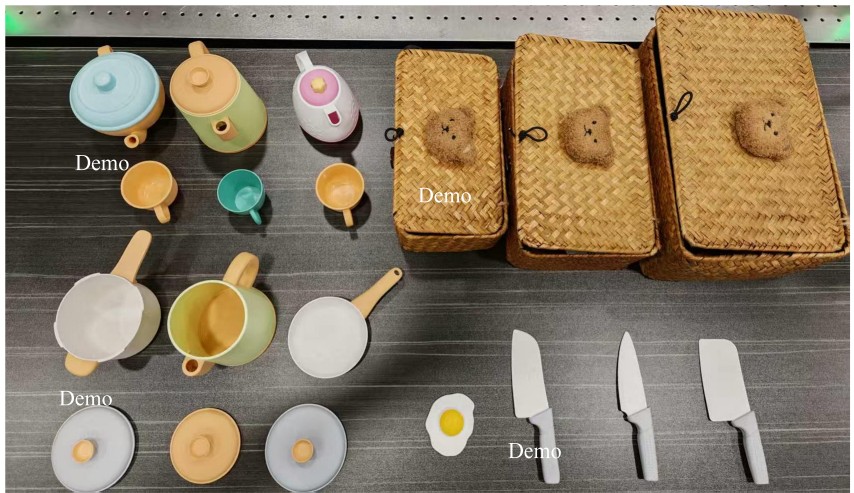

Figure 8: The diverse set of test objects

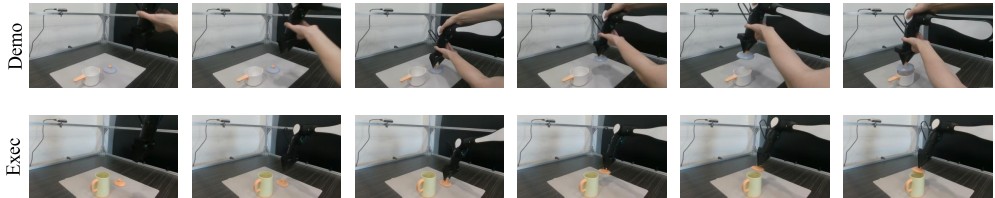

Figure 9: Cross-Category Generalization

3. **Shape:** We replace the demonstration objects with semantically similar counterparts that differ in appearance and geometry, and then randomize the scene according to the "Layout-Easy" setting.

**Inter-Category Generalization**    For example, given a demonstration of placing a lid onto a pot, our policy can successfully adapt to place a lid onto a novel teapot. This result is visualized in Fig. 9.

**Interaction Following**    To test the nuanced understanding of our policy, we provided two demonstrations of the **Pour water** task that *differed only in the grasping style*. Our model not only succeeded at the task but also precisely replicated the distinct interaction style from each demonstration (See Fig. 10).

## C    LARGE LANGUAGE MODEL USAGE

In the development of this project, we employed large language models (LLMs) as supportive tools in two key areas. First, they were used to assist in writing by helping to refine the clarity and flow of this manuscript. Second, in the software implementation phase, LLMs were utilized to identify and suggest fixes for code bugs, as well as to generate initial templates for standard software modules. All outputs were critically reviewed and validated by the authors.

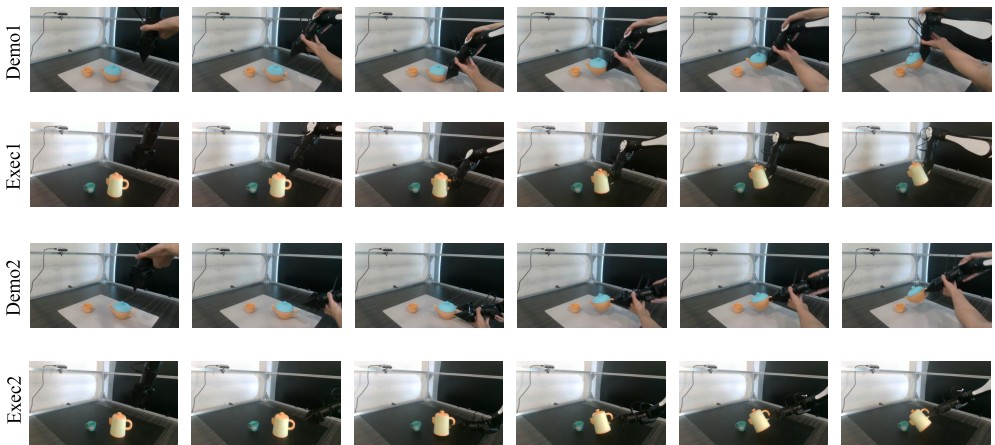

Figure 10: Interaction following shown in the demonstrations.

