# OpenReview forum: "MatchingPolicy: Correspondence-aware Policy For Cross-object In-context Learning"
_ICLR.cc/2026/Conference — Submitted to ICLR 2026_

### Official Review · Reviewer_Dys6 · 2025-10-26

**Soundness:** 3
**Presentation:** 2
**Contribution:** 2
**Rating:** 4
**Confidence:** 4

**Summary:**

The paper proposes **MatchingPolicy** for in-context imitation learning. The authors hypothesize that this failure stems from forcing a single model to simultaneously infer cross-scene correspondences and adapt robot actions. MatchingPolicy consists of a two-stage matching algorithm that leverages off-the-shelf Vision Foundation Models to establish dense semantic correspondences between demonstration frames and the current scene, and a correspondence-aware graph-based diffusion policy that takes these explicit correspondences as input and generates actions. The paper demonstrates that the method outperforms the SOTA baseline (Instant Policy) in simulation (RLBench) and real-world manipulation tasks, and achieves generalization to unseen objects and novel layouts.

**Strengths:**

1. **Clear motivation**: The paper hypothesizes the problem of existing methods and proposes a corresponding solution.
2. **Strong empirical result**: The proposed method significantly outperforms the existing method both in simulation and real-world. The real-world experiment addresses the concern of sim-to-real transfer and demonstrates the potential of further applications.
3. **Novel approach**: The method of combining semantic correspondences and graph-based in-context imitation learning is novel.

**Weaknesses:**

1. **Simulated Data and Simple Primitives**: The policy is trained on a synthetic dataset generated with privileged information (object meshes, viewpoints) in simulations. The tasks themselves are composed of simple, predefined motion primitives. While the model can be fine-tuned, the experiments only show this on simulated data. This raises concerns about scalability to more complex, unstructured real-world tasks that cannot be easily decomposed into simple primitives and do not have privileged perception.

2. **Limitations of Keypoint Representation for Complex Tasks**: As noted in the paper's limitations, the focus is on short-horizon tasks. The abstraction to keypoint correspondences, while powerful for geometric alignment, may fundamentally limit the policy's ability to understand and execute longer, more complex tasks. Such tasks may require reasoning about abstract goals, tool use, or sequential logic that cannot be captured by point-to-point matching. Especially when in-context learning, the contexts are often OOD of the training set.

3. **Constrained Hardware Assumptions**: The high-performance "two-stage matching" algorithm relies on a specific multi-camera setup (three cameras with 360 coverage) to resolve pose ambiguity. This is a significant hardware constraint that may limit the method's applicability outside of structured, tabletop environments and makes it less general than a single-camera approach.

4. **Clarity and Organization**: The paper's narrative structure was somewhat difficult to follow. For instance, the policy architecture is detailed before its primary inputs (the correspondences) are fully introduced and explained. The subsections felt disconnected at times, requiring the reader to jump back and forth to construct a complete picture of the method's pipeline.

**Questions:**

1. Could the authors provide qualitative visualizations for the matching results referenced in Figure 5? The paper claims "more accurate semantic correspondence," and visualizing this comparison would be very beneficial. Furthermore, can the authors elaborate on why their matching stage is superior? The distinction of performing projection before or after correspondence matching does not seem so critical.

2. There is prior work [1] using rejection sampling for more robust visual (DINO) and geometry (FPFH) features for keypoint matching. They also achieve cross-object and cross-layout generalization in a low-data regime. How does this keypoint proposal method compare to your method?

3. Given the current task complexity and horizon, is in-context learning really needed? How well will a multitask, keypoint-conditioned policy [1] perform on these tasks in terms of success rate and data efficiency? Or even optimization-based methods [2] with foundation models (VLMs, VFMs) in a zero-shot manner?

4. The paper hypothesizes two problems with existing in-context learning methods. Are there references or experiments to support them?

[1] Fang, Xiaolin, et al. "KALM: Keypoint Abstraction Using Large Models for Object-Relative Imitation Learning." 2025 IEEE International Conference on Robotics and Automation (ICRA). IEEE, 2025.

[2] Huang, Wenlong, et al. "Rekep: Spatio-temporal reasoning of relational keypoint constraints for robotic manipulation." Conference on Robot Learning (CoRL). 2024.

---

> ### Author Response · Authors · 2025-11-23
> **Response to Reviewer Dys6 [1/2]**
>
> Dear Reviewer Dys6,
>
> We sincerely appreciate your thoughtful feedback. Please find our responses to your questions below.
>
> **Simulated Data, Simple Primitives may not cover complex, unstructured tasks (W1)**
>
> > “The policy is trained on a synthetic dataset generated with privileged information (object meshes, viewpoints) in simulations. The tasks themselves are composed of simple, predefined motion primitives. While the model can be fine-tuned, the experiments only show this on simulated data. This raises concerns about scalability to more complex, unstructured real-world tasks that cannot be easily decomposed into simple primitives and do not have privileged perception.”
> >
>
> **Answer:**
> It is noteworthy that most of the complex tasks in our evaluation (in both simulation and real-world environments) are structurally distinct from those used in training. Remarkably, MatchingPolicy can generalize to real-world tasks with environmental variations **without any privileged information or fine-tuning**.
>
> We address the reviewer's concerns by clarifying our inference setup and the distinction between our training primitives and evaluation tasks.
>
> **1. No Privileged Information Required for Inference:**
> MatchingPolicy is designed to be agnostic to the source of the correspondences. While we utilize privileged information (e.g., object meshes) during training to generate ground-truth supervision, our policy can operate without these privileges at test time. In real-world deployment, the system relies solely on RGB-D inputs and large vision models (DINO / SAM) to extract correspondences. Our real-world success (Section 4.3) confirms that the policy is robust to the domain gap between "perfect simulation labels" and "noisy real-world perception."
>
> **2. Generalization to Non-Primitive, Complex Tasks:**
> The randomized, primitive-based tasks are a scaffold for learning "how to dynamically compose actions to achieve a demonstrated goal based on visual correspondences." The model does not simply replay a pre-defined "push" or "pull." Instead, it learns the underlying principle of aligning its actions with the progression of correspondences shown in the demonstration.
>
> **Visualization of matching results & Why is your matching stage superior? (Q1 & Q2)**
>
> > “Could the authors provide qualitative visualizations for the matching results referenced in Figure 5? ...Furthermore, can the authors elaborate on why their matching stage is superior? The distinction of performing projection before or after correspondence matching does not seem so critical.“
> >
>
> > “There is prior work [1] using rejection sampling for more robust visual (DINO) and geometry (FPFH) features for keypoint matching. They also achieve cross-object and cross-layout generalization in a low-data regime. How does this keypoint proposal method compare to your method?”
> >
>
> **Answer:**
> We found that matching before projection is critical because projecting features from different views into a unified 3D space disrupts their inherent 2D geometry. Specifically, points that are neighbors in the current 2D observation can be incorrectly matched to distant points from a demonstration view based purely on semantic similarity, leading to numerous mismatches.
>
> While KALM [1] introduced a geometric correspondence method to mitigate this, we found that a naive 3D matching algorithm combined with this approach still produces numerous mismatches, particularly under large layout variations where object symmetry becomes a confounding factor. This analysis is supported by visual correspondence results, available on [our anonymous website](https://matchingpolicy.github.io/).
>
> **Limitations of Keypoint Representation for complex, long-horizon tasks. (W2)**
>
> > “As noted in the paper's limitations, the focus is on short-horizon tasks. The abstraction to keypoint correspondences, while powerful for geometric alignment, may fundamentally limit the policy's ability to understand and execute longer, more complex tasks. Such tasks may require reasoning about abstract goals, tool use, or sequential logic that cannot be captured by point-to-point matching. Especially when in-context learning, the contexts are often OOD of the training set.”
> >
>
> **Answer:**
> We fully agree that MatchingPolicy's keypoint-based abstraction is not specifically designed for long-horizon reasoning. Our envisioned solution is to take it as a plug-in module for the robust low-level executor within a hierarchical architecture. In this architecture, a high-level planner (e.g., a VLM) would decompose long-horizon tasks into sequences of geometric sub-tasks. MatchingPolicy would then execute these sub-tasks with precision. The division of labor combines the VLM's semantic reasoning with MatchingPolicy's precise motion control, addressing long-horizon planning and improving OOD generalization by recombining primitive skills, without needing end-to-end training.

---

> ### Author Response · Authors · 2025-11-23
> **Response to Reviewer Dys6 [2/2]**
>
> **Is In-Context Learning needed? Why not Multi-task or Zero-shot? (Q3)**
>
> > “Given the current task complexity and horizon, is in-context learning really needed? How well will a multitask, keypoint-conditioned policy [1] perform on these tasks in terms of success rate and data efficiency?"
> >
>
> **Answer:**
> We position in-context learning as a trade-off between ease of use (compared to task-specific fine-tuning) and performance (compared to zero-shot learning). To validate this, a comparison with a zero-shot generalist policy is needed. However, since no similar keypoints-conditional policy exists (KALM is designed for a single task), we adapted a representative Vision-Language Action model (VLA) [2]. We evaluated this VLA on six unseen tasks under zero-shot and few-shot settings.
>
> As shown in our general response, the VLA struggled to generalize, achieving near-zero success in the zero-shot setting. This is expected due to the significant distribution shift, as our tasks are out-of-distribution relative to its pre-training data. Crucially, even when provided with 20 demonstrations, this VLA was outperformed by our MatchingPolicy using only 2 demonstrations. This result strongly illustrates the necessity and efficiency of in-context learning for rapid adaptation to novel tasks.
>
> > "Or even optimization-based methods [2] with foundation models (VLMs, VFMs) in a zero-shot manner?”
> >
>
> **Answer:**
> As mentioned by the Reviewer, while VLM-based, optimization-driven methods [2] could be applied to our setting, their performance is heavily reliant on carefully chosen, human-annotated keypoints. In contrast, our MatchingPolicy operates effectively with non-specific points.
>
> **Evidence supporting the hypothesis of ``two problems'' (Implicit Matching & Simultaneous Burden). (Q4)**
>
> > “The paper hypothesizes two problems with existing in-context learning methods. Are there references or experiments to support them?”
> >
>
> **Answer:**
> We posit that correspondence extraction and motion transfer are two essential tasks for in-context learning to predict accurate robot actions. This structure is echoed in explicit one-shot imitation methods [8,9,10], which decompose the problem into two analogous stages.
>
> Although end-to-end methods do not maintain this explicit separation, we observe symptoms of correspondence mismatch in our real-world experiments with InstantPolicy. This is demonstrated by the "put lid" task in a "layout-hard" setting, where the robot mistakes the pot for the lid. This error in correspondence causes the policy to train to grasp the wrong object, resulting in the low success rate shown in our comparative experiments (Table 2). We have upload the failure case on [our project website.](https://matchingpolicy.github.io/)
>
> **Constrained Hardware Assumptions (3-camera setup). (W3)**
>
> > “The high-performance "two-stage matching" algorithm relies on a specific multi-camera setup (three cameras with 360 coverage) to resolve pose ambiguity. This is a significant hardware constraint that may limit the method's applicability outside of structured, tabletop environments and makes it less general than a single-camera approach.”
> >
>
> **Answer:**
> Thank you for your question. Please refer to our General Response.
>
> **Clarity and Organization. (W4)**
>
> > “The paper's narrative structure was somewhat difficult to follow. For instance, the policy architecture is detailed before its primary inputs (the correspondences) are fully introduced and explained.“
> >
>
> **Answer:**
> Thank you for this feedback. We have restructured the paper by adding a definition of "correspondences" in the overview to improve readability. Please refer to the Section 3 in our revised paper.
>
>
>
> **Reference**
>
> - [1] KALM: Keypoint Abstraction using Large Models for Object-Relative Imitation Learning, CORL 2024
> - [2] RDT-1B: a Diffusion Foundation Model for Bimanual Manipulation, ICLR 2025
> - [3] Instant Policy: In-Context Imitation Learning via Graph Diffusion, ICLR 2025
> - [4] UniGarmentManip: A Unified Framework for Category-Level Garment Manipulation via Dense Visual Correspondence, CVPR 2024
> - [5] Keypoint Action Tokens Enable In-Context Imitation Learning in Robotics, RSS 2024
> - [6] RICL: Adding In-Context Adaptability to Pre-Trained Vision-Language-Action Models, CoRL 2025
> - [7] REGENT: A Retrieval-Augmented Generalist Agent That Can Act In-Context In New Environments, ICLR 2025
> - [8] Heppert, Nick, et al. "Ditto: Demonstration imitation by trajectory transformation." IROS 2024
> - [9] Zhu, Yifeng, et al. "Vision-based manipulation from single human video with open-world object graphs." arXiv 2024.
> - [10] Tang, Chao, et al. "Mimicfunc: Imitating tool manipulation from a single human video via functional correspondence." CoRL 2025.
>
> *Once again, thank you for your valuable time and effort in reviewing our work.*
>
> Authors of Submission2902

---

### Official Review · Reviewer_ScEQ · 2025-10-28

**Soundness:** 3
**Presentation:** 2
**Contribution:** 2
**Rating:** 4
**Confidence:** 3

**Summary:**

MatchingPolicy proposes a novel in-context imitation learning method. They isolate a challenge of in-context learning to the combined difficulty of inferring correspondences between the demo and current scene, and then adapting new actions to the scene. To address this challenge, MatchingPolicy decomposes the correspondence matching between demo and current scene from the action adaptation part. First, a group of representative points from the point cloud are selected, which can be done by taking the center of the semantic segmentation of an object. This allows matching temporally (across different frames) with off-the-shelf models, and also matching between the demo frames and current frames. The actions can extracted from the gripper keypoints. The model is trained using a procedurally generated dataset (pretraining) and a human collected dataset in sim and real (finetuning). The model is a graph-based diffusion model.

**Strengths:**

1. The introduction presents the idea well.
2. The experiments section is fairly thorough in describing the different tasks.
3. The test-time planning ablation seems orthogonal to the thesis of the paper, but it is interesting that Overlap outperforms half so much.

**Weaknesses:**

1. Figure 2 is a bit confusing. More details can be added to the caption explaining the orange and blue lines? Even in the text, I was a bit confused by exactly what the lines meant.
2. I feel like additional baselines are missing.
3. The authors could probably explain more about how InstantPolicy is different from MatchingPolicy, especially given that it is the only other baseline.

**Questions:**

1. DINO is mentioned in the intro and never mentioned again. How would you use DINO in this decoupled in-context learning design?
2. How much training data is generated?
3. How much finetuning data is collected? How diverse and semantically different is this finetuning dataset?
4. The selection of points is not super clear to me. How exactly are the points chosen from the pointcloud?
5. How many diffusion steps are used for the policy during training and inference?

---

> ### Author Response · Authors · 2025-11-23
> **Response to Reviewer ScEQ [1/2]**
>
> Dear Reviewer ScEQ,
>
> We sincerely appreciate your thoughtful feedback. Please find our responses to your questions below.
>
> **"How is InstantPolicy different from MatchingPolicy?" (W3)**
>
> MatchingPolicy differs from InstantPolicy in three fundamental aspects:
>
> - **Point Selection:** MatchingPolicy takes explicitly matched 3d correspondences across frames as input, whereas InstantPolicy relies on points independently selected via Farthest Point Sampling (FPS) without temporal alignment.
> - **Graph Structure:** MatchingPolicy constructs *direct edges* between these matched points to leverage strong spatiotemporal priors. In contrast, InstantPolicy lacks these explicit connections and treats point clouds more independently.
> - **Training Utilization:** MatchingPolicy is explicitly optimized to utilize correspondence history for action prediction, whereas InstantPolicy learns solely from implicit geometric features.
>
>
> **"I feel like additional baselines are missing." (W2)**
>
> We prioritized InstantPolicy [3] in our initial submission as it serves as the most direct counterpart in the strict in-context imitation setting (enabling instant adaptation from 1-2 demos).
>
> However, to address the reviewer's concern and place our method in a broader context, we have expanded our evaluation to include representative methods from more paradigms: KAT [1] (LLM-based ICL) and RDT-1B [2] (VLA Foundation Model). Although these methods operate under different assumptions, we established a unified evaluation protocol to ensure a rigorous and fair comparison. Please refer to the General Response for the detailed results and analysis.
>
>
> **"DINO is mentioned in the intro and never mentioned again. How is it used?" (Q1)**
>
> DINOv2 [4] serves as the visual descriptors for our two-stage matching (Section 3.3). Specifically, DINO features are utilized to:
>
> 1. Calculate global similarity for retrieving the optimal demonstration viewpoints (View Matching).
> 2. Compute pixel-wise correlations to establish precise correspondences between the demonstration and the current scene (Point Matching).
>
> We further note that our framework is agnostic to the specific backbone and is compatible with other dense descriptors such as SD-DINO [5] and DIFT [6].
>
>
>  **"How much training data is generated?" (Q2)**
>
> We generated a total of 52k pseudo-task groups, amounting to $\sim$156k individual trajectories (3 randomized rollouts per group). The detailed breakdown by motion primitive is provided below.
>
> | Primitive Type | Pick-and-Place | Open-and-Close | Grasping | Push-Pull | Path Following | **Total** |
> | --- | --- | --- | --- | --- | --- | --- |
> | **Task Groups** | 10k | 10k | 6k | 6k | 20k | **52k** |
>
> *This breakdown has been added to Appendix A in the revised manuscript.*
>
>
>  **"How much finetuning data is collected? How diverse is it?" (Q3)**
>
> For the simulated experiments, we provide 20 demonstrations for each fine-tuning task to fine-tune the pre-trained models. For the real-world experiments, we evaluate the models without any fine-tuning.

---

> ### Author Response · Authors · 2025-11-23
> **Response to Reviewer ScEQ [2/2]**
>
> **"Figure 2 is a bit confusing. More details can be added to the caption explaining the orange and blue lines? Even in the text, I was a bit confused by exactly what the lines meant." (W1)**
>
> We apologize for the lack of clarity. We have substantially revised **Figure 2** and its caption.
>
> In the graph, lines (excluding the red ones) represent edges for propagating information, with the key distinction being that a **solid** line denotes a single relational edge between two corresponding nodes of two frames, while a **dotted** line represents a group of point-to-point edges for each pair of corresponding nodes. Specifically:
>
> - An **orange solid line** is a relational edge between the matched gripper or scene nodes of two frames.
> - An **orange dotted line** is a group of point-to-point edges for each corresponding node pair.
> - A **blue solid line** is a relational edge between two matched gripper nodes of two frames.
> - A **blue dotted line** is a group of point-to-point edges for each pair of corresponding gripper nodes.
>
> These clarifications have also been integrated into the updated caption and relevant descriptions in the revised manuscript.
>
>
> **"The selection of points is not super clear to me. How exactly are the points chosen from the pointcloud?" (Q4)**
>
> We select points from the current observation's point cloud based on two criteria: first, each point must have a semantically matched correspondence visible across most demonstration frames; second, the final set must be both representative of the scene and sufficiently random for robustness.
>
> To this end, we employ a two-step selection process. First, for every point in the current observation, we compute its correspondences in the demonstration point clouds using template matching during training (Section 3.2) or two-stage matching during execution (Section 3.3). Points that meet the first criterion are retained as candidates. Second, we apply Farthest Point Sampling (FPS) to these candidates to select a final, spatially distributed set of points. Their corresponding points in the demonstration frames form the representative set for the demonstrations.
>
>  **"How many diffusion steps are used?" (Q5)**
>
> We employ a DDIM scheduler with a squared-cosine beta schedule. During training, the noise schedule is discretized into $T=100$ timesteps. For inference, we use a reduced sampling trajectory of $T=4$ steps to ensure real-time performance.
>
>
> **References**
>
> - [1] Keypoint Action Tokens Enable In-Context Imitation Learning in Robotics, RSS 2024
> - [2] RDT-1B: a Diffusion Foundation Model for Bimanual Manipulation, ICLR 2025
> - [3] Instant Policy: In-Context Imitation Learning via Graph Diffusion, ICLR 2025
> - [4] DINOv2: Learning Robust Visual Features without Supervision, TMLR
> - [5] A Tale of Two Features: Stable Diffusion Complements DINO for Zero-Shot Semantic Correspondence, NeurIPS 2023
> - [6] Emergent Correspondence from Image Diffusion, NeurIPS 2023
>
> *Once again, thank you for your valuable time and effort in reviewing our work.*
>
> Authors of Submission2902

---

### Official Review · Reviewer_iphX · 2025-10-31

**Soundness:** 2
**Presentation:** 2
**Contribution:** 2
**Rating:** 4
**Confidence:** 3

**Summary:**

This paper proposes MatchingPolicy, an approach for in-context imitation learning for robotic manipulation that decouples demonstration-to-scene matching with a correspondence-aware graph-based diffusion policy. To deploy in the real world, the authors introduce a two-stage feature matching algorithm using VLMs for real-time correspondence extraction. They identify the optimal alignment between the testing and demonstration view, perform 2d feature point matching, project into 3d point cloud, and then apply farthest point sampling to obtain representative keypoints with encoded geometric features for selected points in local neighborhoods.

**Strengths:**

- This paper studies an important topic of how to leverage in-context learning for robotic policy learning. The proposed approach seems to drastically outperform the prior baseline on in-context imitation learning [Vosylius and Johns, 2024] in unseen tasks, demonstrating strong generalization. The real-world results are promising, given that the policy is only provided 1-2 in-context demonstrations and trained entirely in simulation. Particularly as the simulation tasks are [pick-and-place, grasping, push-pull, random path following], but the experimental results show that the policy is able to perform pouring, opening, and cutting tasks in the real-world without additional fine-tuning.

- The paper is generally well-written and easy to follow.

**Weaknesses:**

- For Appendix B Figure 10, it looks like the demonstrations show a human hand guiding the robot for kinesthetic teaching. In that case, I may expect that the performance gain from MatchingPolicy comes from effectively masking out the human demonstrator’s hand in the visual information for the policy. Given a teleoperation setup such as ALOHA or GELLO to enable directly collecting demonstrations in the scene without interference, how well would the baseline InstantPolicy perform?

- Only a single in-context imitation learning baseline (InstantPolicy) is used for comparison. How does MatchingPolicy fare against zero-shot approaches that use VLMs to extract representations for real-world robot manipulation, such as ReKep [1] or Voxposer [2]?

- Regarding simulation results on RLBench, a standard robotics benchmark, it would be helpful to compare against a wider range of baselines beyond a single in-context imitation learning baseline, particularly on “Unseen Tasks”. How does MatchingPolicy perform compared to VLAs?

- The real-world evaluation tasks only include at most two objects in the scene. How well does MatchingPolicy work when there are other objects in the scene that may be distractors? How about when there are more than two objects involved in a manipulation interaction? For instance, picking up a plate with multiple utensils on it.

- The proposed approach requires 3 cameras in a 360deg setup to capture the entire 3d scene and resolve pose ambiguity.

[1] https://arxiv.org/abs/2409.01652

[2] https://arxiv.org/abs/2307.05973

**Questions:**

Figure 4: Where are the visualizations of the “Cut Egg” task?

How does MatchingPolicy perform with more than 2 in-context demonstrations?

[minor]

L199-201: This sentence is difficult to understand, what are “in demonstration frames”?

---

> ### Author Response · Authors · 2025-11-23
> **Response to Reviewer iphX [1/2]**
>
> Dear Reviewer iphX,
>
> We sincerely appreciate your thoughtful feedback. Please find our responses to your questions below.
>
> **Performance with more than 2 in-context demonstrations. (Q2)**
>
> > “How does MatchingPolicy perform with more than 2 in-context demonstrations?”
> >
>
> We highlight our approach's exceptional data efficiency—achieving high success rates with only one or two demonstrations—as a key advantage over prior methods. Although our framework accepts a variable number of demonstrations, performance typically saturates at just N=1 or 2. We posit that this is because one or two demonstrations already provide sufficient information for the task, and further additions yield negligible gains at increased computational cost.
> This efficiency is further highlighted by our comparison with VLA methods (see General Response). MatchingPolicy outperforms the VLA even when the latter uses 20 demonstrations with fine-tuning.
>
> **Additional Comparison with VLAs and VLM approaches. (W2 & W3)**
>
> > “Only a single in-context imitation learning baseline (InstantPolicy) is used for comparison. How does MatchingPolicy fare against zero-shot approaches that use VLMs to extract representations for real-world robot manipulation, such as ReKep [1] or Voxposer [2]"
> >
> >
> > Regarding simulation results on RLBench, a standard robotics benchmark, it would be helpful to compare against a wider range of baselines beyond a single in-context imitation learning baseline, particularly on “Unseen Tasks”. How does MatchingPolicy perform compared to VLAs?”
> >
>
> In response to the reviewer's suggestion, we expanded our evaluation to include comparisons against representative approaches: a state-of-the-art VLA model (RDT-1B [3]) and an LLM-based method (KAT [5]). Please refer to the General Response for detailed tables.
>
> **1. Comparison with VLAs:** We evaluated RDT-1B on six unseen tasks. As detailed in the General Response, we observe two findings:
>
> - **Zero-shot Transfer:** The VLA struggles to generalize (near-zero success) in the zero-shot setting (no demonstrations). This is an expected result due to distribution shifts, as these unseen tasks are out-of-distribution regarding the VLA's pre-training data.
> - **Data Efficiency:** Strikingly, MatchingPolicy (using 2 demos) outperforms the VLA even when the latter is fine-tuned on 20 demonstrations. This demonstrates the superior data efficiency of our explicit correspondence modeling compared to VLA fine-tuning.
>
> **2. Comparison with VLM Approaches:**
> Voxposer [2] generates motion constraints via 3D voxel maps but can struggle with the precise, contact-rich tasks we target.
> ReKep [1] utilizes keypoints to formulate constraints for motion optimization but relies heavily on human-annotated points for high performance; its efficacy drops significantly with automatically selected points, suggesting a limitation in VLMs' spatial reasoning.
> In contrast, our method operates with non-specific points, eliminating the need for human annotation. For a more relevant and fair comparison, we therefore evaluate against KAT [5], an LLM-based in-context learning method that similarly requires no human-annotated keypoints. Our results (Table 1) serve as a strong benchmark: even when provided with visual in-context examples, KAT achieved low success on precision tasks like "Slide Buzzer."
>
> **Robustness to distractors and multi-object interactions. (W4)**
>
> > “The real-world evaluation tasks only include at most two objects in the scene. How well does MatchingPolicy work when there are other objects in the scene that may be distractors? How about when there are more than two objects involved in a manipulation interaction? For instance, picking up a plate with multiple utensils on it.”
> >
>
> By decoupling visual correspondence extraction from policy execution, we can utilize off-the-shelf segmentation models [4] to explicitly filter out distractors.
> This allows our approach to function in cluttered environments.
> To verify this claim, we constructed a real-world scenario with multi-object interactions and distractors. Our method successfully completed the task using a single demonstration. Please visit [our anonymous website](https://matchingpolicy.github.io/) for more details.

---

> ### Author Response · Authors · 2025-11-23
> **Response to Reviewer iphX [2/2]**
>
> **Kinesthetic teaching and hand masking. How would InstantPolicy perform with teleoperation? (W1)**
>
> > “For Appendix B Figure 10, it looks like the demonstrations show a human hand guiding the robot for kinesthetic teaching. In that case, I may expect that the performance gain from MatchingPolicy comes from effectively masking out the human demonstrator’s hand in the visual information for the policy. Given a teleoperation setup such as ALOHA or GELLO to enable directly collecting demonstrations in the scene without interference, how well would the baseline InstantPolicy perform?”
> >
>
> We would like to clarify that the hand point cloud is explicitly masked out for both InstantPolicy and MatchingPolicy before being fed into the policy. Thus, regardless of whether the data is collected via kinesthetic teaching or teleoperation (e.g., ALOHA), the model inputs are standardized as clean objects' point cloud, ensuring a fair comparison.
> Given this consistent input, the performance gain of our method does not stem from data cleaning, but from the core algorithm: MatchingPolicy explicitly models visual correspondence from RGB-D data using off-the-shelf foundation models, whereas InstantPolicy infer implicit correspondence from random selected points with only geometric features.
>
> **The Concern about Camera Constraints for Method universality. (W5)**
>
> > “The proposed approach requires 3 cameras in a 360deg setup to capture the entire 3d scene and resolve pose ambiguity.”
> >
>
> Thanks for your question. Please refer to the General Response.
>
> **Visualization of "Cut Egg". (Q1)**
>
> > “Figure 4: Where are the visualizations of the “Cut Egg” task?”
> >
>
> The visualization of the "cut egg" is not included in the main paper due to space constraints; however, a schematic of the task is provided in Appendix, Fig. 7. The corresponding video has also been updated on [our anonymized website](https://matchingpolicy.github.io/) for reference.
>
> **What are “in demonstration frames” in L199-201. (Q3)**
>
> We added a defination of "frame" in the Problem Formulation in Section 3 in our revised manuscript.
>
> **Reference**
>
> [1] ReKep : Spatio-Temporal Reasoning of Relational Keypoint Constraints for Robotic Manipulation, CoRL 2024
> [2] VoxPoser: Composable 3D Value Maps for Robotic Manipulation with Language Models, CoRL 2023
> [3] RDT-1B: a Diffusion Foundation Model for Bimanual Manipulation, ICLR 2025
> [4] SAM 2: Segment Anything in Images and Videos, ICLR 2025
> [5] Keypoint Action Tokens Enable In-Context Imitation Learning in Robotics, RSS 2024
>
> *Once again, thank you for your valuable time and effort in reviewing our work.*
>
> Authors of Submission2902

---

> > ### Comment · Reviewer_iphX · 2025-11-27
> >
> > Thank you for the extensive responses and updated experimental results. I have further concerns regarding the number of demos used for in-context learning, but I will increase my score to 6 in a show of good faith.
> >
> > ---
> >
> > > Although our framework accepts a variable number of demonstrations, performance typically saturates at just N=1 or 2. We posit that this is because one or two demonstrations already provide sufficient information for the task, and further additions yield negligible gains at increased computational cost.
> >
> > So MatchingPolicy performance does not improve with additional demos? Why might this be the case? In LLMs, in-context learning performance generally scales with the number of in-context demos. In Table 2, performance has not yet saturated on tasks such as "Unplug charger". I think it would be helpful to report MatchingPolicy performance with $N=1$ thru $N=4$ or $5$ demos to get a sense of potential scalability. See Table 4 in InstantPolicy [Vosylius and Johns, 2025] for an example of this type of evaluation.
> >
> > > We would like to clarify that the hand point cloud is explicitly masked out for both InstantPolicy and MatchingPolicy before being fed into the policy. Thus, regardless of whether the data is collected via kinesthetic teaching or teleoperation (e.g., ALOHA), the model inputs are standardized as clean objects' point cloud, ensuring a fair comparison.
> >
> > How is this done with VLAs that use RGB image inputs? I assume you use the same demonstration data for the latest VLA comparisons in the rebuttal.

---

> > > ### Author Response · Authors · 2025-12-03
> > >
> > > Thank you for raising your score and providing additional comments! Below are our responses.
> > >
> > > > So MatchingPolicy performance does not improve with additional demos? Why might this be the case? In LLMs, in-context learning performance generally scales with the number of in-context demos. In Table 2, performance has not yet saturated on tasks such as "Unplug charger". I think it would be helpful to report MatchingPolicy performance with N=1
> > > thru N=4 or N=5 demos to get a sense of potential scalability. See Table 4 in InstantPolicy [Vosylius and Johns, 2025] for an example of this type of evaluation.
> > > >
> > >
> > >  We evaluated MatchingPolicy with varying demonstration sizes (N=1, 2, 4). We observed that naively stacking demonstrations (N=4) does not strictly improve performance. However, we found that scalability can be achieved through integreting a selection-based strategy. When we use the Retrieval-Augmented Generation (RAG) similar to [1] to select the Top-2 most visually relevant demonstrations, the performance consistently improves beyond the N=2 baseline.
> > >
> > > | Unseen Task from RLBench | demo 1 | demo 2 | demo 4 (naive) | demo 4 (RAG) |
> > > | --- | --- | --- | --- | --- |
> > > | Plate Out | 0.92 | 0.96 | 0.97 | 0.98 |
> > > | Slide Buzzer | 0.63 | 0.71 | 0.75 | 0.80 |
> > > | Toilet Seat Up | 0.98 | 1.00 | 1.00 | 1.00 |
> > > | Meat off grill | 0.72 | 0.77 | 0.72 | 0.81 |
> > > | Pick up Cup | 0.90 | 0.94 | 0.86 | 0.96 |
> > > | Unplug charger | 0.41 | 0.46 | 0.35 | 0.49 |
> > >
> > > We will investigate the underlying reason for the interference observed in naive stacking and work to enhance the robustness of MatchingPolicy to redundant contexts in our future work.
> > >
> > > > How is this done with VLAs that use RGB image inputs? I assume you use the same demonstration data for the latest VLA comparisons in the rebuttal.
> > > >
> > >
> > > All baselines (including VLAs) use identical demonstration data (action labels, task sequences) and RLBench’s distraction-free tabletop setup. For VLAs, we input raw RGB images as their original training format, instead of segmented point clouds in MatchingPolicy and InstantPolicy, to best preserve their pretraining capabilities.
> > >
> > > [1] RICL: Adding In-Context Adaptability to Pre-Trained Vision-Language-Action Models, CoRL 2025
> > >
> > > Thank you again for your response. We welcome any further advice or feedback you might offer.
> > >
> > > Authors of Submission2902

---

### Official Review · Reviewer_ejwA · 2025-11-03

**Soundness:** 3
**Presentation:** 3
**Contribution:** 2
**Rating:** 4
**Confidence:** 3

**Summary:**

The paper introduces MatchingPolicy, a new framework for ICIL, that explicitly decouples correspondence extraction from policy learning. Rather than simply throwing the keypoints from demonstrations and the ongoing rollout into a GNN diffusion policy, the proposed method explicitly connects similar nodes (grippers and scene nodes in ongoing rollouts to those in the demos) before feeding it into the GNN diffusion policy. This explicit representation connexting the state and demonstrations makes it easier for the GNN diffusion policy to predict relevant actions. These direct correspondences are obtained using off-the-shelf models. The method might be seen as InstantPolicy + more explicit representations. This results in improved generalization capabilities to unseen tasks in both sim (RLBench) and real (with novel objects, layouts, etc).

**Strengths:**

Explicit representations were very essential to meta learning in supervised setting, including but not limited to Matching Networks, ProtoNets, and SimpleShot. I am very excited by a similar approach to meta policy learning.

The results on generalization to new objects, new layouts, and in "cross-category" generalization are very interesting.

The significant improvements over instant-policy are very nice to observe. The ablations are helpful in establishing the effectiveness of the matching procedure.

**Weaknesses:**

Missing related work with similar (or atleast relevant) high-level ideas: A few highly relevant papers in in-context imitation learning are missing from the related work, and possibly from the baselines. The following papers [1, 2, 3] focus on generalization to novel objects, tasks, and environments. In fact, both RICL and REGENT follow a RAG approach that also decouples matching and acting.

[1] RICL: Adding In-Context Adaptability to Pre-Trained Vision-Language-Action Models, CoRL 2025

[2] REGENT: A Retrieval-Augmented Generalist Agent That Can Act In-Context In New Environments, ICLR 2025

[3] Generalization to New Sequential Decision Making Tasks with In-Context Learning, ICML 2024

Are other baselines in the keypoint ICIL methods such as KAT [4] necessary in this context? If so, they are currently not mentioned in the paper.

[4] Keypoint Action Tokens Enable In-Context Imitation Learning in Robotics, RSS 2024

**Questions:**

Please see weaknesses

---

> ### Author Response · Authors · 2025-11-23
> **Response to Reviewer ejwA**
>
> Dear Reviewer ejwA,
>
> We sincerely appreciate your thoughtful feedback. Please find our responses to your questions below.
>
> **Missing related work with high-level ideas. (Q1)**
>
> > “Missing related work with similar (or atleast relevant) high-level ideas: A few highly relevant papers in in-context imitation learning are missing from the related work... [1] RICL... [2] REGENT... [3] Generalization to New Sequential Decision Making Tasks... In fact, both RICL and REGENT follow a RAG approach that also decouples matching and acting.”
> >
>
> We thank the reviewer for pointing out this important oversight. These works constitute a complementary set of references [1,2,3] for our related work section. Here, we discuss the key differences between these works and MatchingPolicy.
>
> - **Fine-Grained 3D Correspondence for Action Prediction:** While both approaches decompose in-context learning into "matching" and "action prediction" stages, MatchingPolicy differs by extracting dense, point-wise 3D correspondences between all demonstration states and the current scene. This provides fine-grained spatial information that facilitates the robot's action prediction, moving beyond simply retrieving the most similar states like RICL [1] and REGENT [2].
> - **Sample Efficiency through Specialized Training:** Trained on a specialized, spatially-variant synthetic dataset, MatchingPolicy is explicitly optimized to handle large motion differences using only a limited number of demonstrations. This enables it to adapt to new tasks with high sample efficiency, achieving robust performance with even a single demonstration. This represents a substantial improvement in demonstration efficiency over methods like RICL, which typically require at least 10 demonstrations in their setting.
> - **Potential on Retrieval-Based Scaling:** RICL and REGENT demonstrate the utility of a RAG-based framework for sourcing demonstrations. We consider this a relevant direction for extending our system. Future work could explore integrating such retrieval logic to automatically select reference motions from larger datasets, serving as an effective upstream component to our fine-grained execution.
>
> In the revised manuscript, we have now added a discussion of these works in the Related Work section.
>
> **Comparison with Keypoint ICIL methods like KAT. (Q2)**
>
> > “Are other baselines in the keypoint ICIL methods such as KAT [4] necessary in this context? If so, they are currently not mentioned in the paper. [4] Keypoint Action Tokens Enable In-Context Imitation Learning in Robotics, RSS 2024”
> >
>
> We agree with the reviewer's observation that KAT [4] addresses a similar challenge as our method. In response, we refined our description of KAT in the revised manuscript and added a comprehensive comparison on six unseen RLBench tasks, as detailed in the General Response.
>
> Notably, results show that MatchingPolicy with 2 demos consistently outperforms KAT even when the latter uses 10 demos. KAT's performance is constrained by two primary factors:
>
> - **Spatial Grounding:** The underlying LLM lacks the general knowledge to understand spatial relationships between demonstrations and the current scene, leading to inaccurate and inefficient action inference.
> - **Open-loop Execution:** KAT operates in an open-loop manner; it predicts the entire action trajectory in advance and executes it step-by-step. This lack of real-time adaptation during execution further limits its performance.
>
> **References**
>
> [1] RICL: Adding In-Context Adaptability to Pre-Trained Vision-Language-Action Models, CoRL 2025
>
> [2] REGENT: A Retrieval-Augmented Generalist Agent That Can Act In-Context In New Environments, ICLR 2025
>
> [3] Generalization to New Sequential Decision Making Tasks with In-Context Learning, ICML 2024
>
> [4] Keypoint Action Tokens Enable In-Context Imitation Learning in Robotics, RSS 2024
>
> *Once again, thank you for your valuable time and effort in reviewing our work.*
>
> Authors of Submission2902

---

### Author Response · Authors · 2025-11-23
**General Response**

We thank the reviewers for their encouraging and insightful comments. We are pleased that they acknowledged the motivation and significance of our work, and we are particularly encouraged that they found our approach to be 'novel' and our empirical results 'strong', 'promising' and ‘interesting’. In this general response, we address the main recurring comments. We provide specific, point-by-point replies to each reviewer in the sections that follow.

**Comparison with Additional Methods (Reviewer ejwA, iphX, ScEQ, Dys6):**

We have added comprehensive comparisons on the RLBench benchmark:

- **LLM-based In-context learning methods (Reviewer ejwA):** We compared our method against KAT [1], a keypoint-based in-context learning method. As shown below, MatchingPolicy outperforms KAT on unseen tasks, achieving higher success rates with only 2 demos than KAT achieves with 10 demos:

| Unseen Task from RLBench | MatchingPolicy (Ours, 2 demos) | KAT (Aligned with Ours, 2 demos) | KAT (Original KAT Setting, 10 demos) |
| --- | --- | --- | --- |
| Plate Out | 0.96 | 0.12 | 0.36 |
| Slide Buzzer | 0.71 | 0.04 | 0.19 |
| Toilet Seat Up | 1.00 | 0.25 | 0.38 |
| Meat off grill | 0.77 | 0.33 | 0.54 |
| Pick up Cup | 0.94 | 0.28 | 0.47 |
| Unplug charger | 0.46 | 0.00 | 0.12 |
- **VLA methods (Reviewer iphX):** We evaluated the VLA paradigm using RDT-1B [2] as a SOTA representative. As shown below, the VLA struggles in zero-shot settings, an expected result given that these tasks are out-of-distribution regarding its pre-training data. Strikingly, MatchingPolicy with 2 demos outperforms the VLA even when the latter is fine-tuned on 20 demonstrations, demonstrating the superior data efficiency of our approach.

| Unseen Task from RLBench | MatchingPolicy (Ours, 2 demos) | VLA (Zero-shot Transfer, 0 demo) | VLA (Task-Specific FT, 20 demos) |
| --- | --- | --- | --- |
| Plate Out | 0.96 | 0.08 | 0.67 |
| Slide Buzzer | 0.71 | 0.00 | 0.47 |
| Toilet Seat Up | 1.00 | 0.12 | 0.69 |
| Meat off grill | 0.77 | 0.05 | 0.73 |
| Pick up Cup | 0.94 | 0.09 | 0.81 |
| Unplug charger | 0.46 | 0.00 | 0.32 |

**Constrained Multi-Camera Settings (Reviewer iphX, Dys6):**

Our method remains effective with a single camera under moderate variation ("Layout-Easy" and "Shape" settings). However, significant layout differences remain a challenge for 2D semantic matching, as descriptors like DINO [4] are sensitive to large perspective shifts; this is why a multi-camera setup is currently used to ensure robust correspondence. Notably, concurrent work [5] has achieved robust matching even under substantial layout differences. Integrating such a module could therefore enable the MatchingPolicy to handle complex, layout-diverse scenes using only a single camera, significantly broadening its applicability.

**Visual Ablation of Different Matching Methods (Reviewer Dys6):**

To validate the accuracy of our two-stage matching approach, we provide a qualitative comparison against a naive 3D matching baseline on our [anonymous project website](https://matchingpolicy.github.io/).

**References**

[1] Keypoint Action Tokens Enable In-Context Imitation Learning in Robotics, RSS 2024

[2] RDT-1B: a Diffusion Foundation Model for Bimanual Manipulation, ICLR 2025

[3] Instant Policy: In-Context Imitation Learning via Graph Diffusion, ICLR 2025

[4] DINOv2: Learning Robust Visual Features without Supervision, TMLR

[5] DENSE SEMANTIC MATCHING WITH VGGT PRIOR, Arxiv:2509

---

### Author Response · Authors · 2025-11-23
**Revised Paper**

## Paper Revision

In response to the reviewers' valuable feedback, we have made the following revisions to the manuscript. Minor typographical errors have been corrected throughout but are not detailed in the list below

- **Related Works (Reviewer ejwA, Dys6):** We expanded our discussion to include retrieval-based ICL methods (RICL, REGENT), keypoint-based ICL methods (KAT) and comparisons with task-specific training methods (KALM), clarifying the unique position of MatchingPolicy as a 3d correspondence-based, in-context learning framework.
- **Structural Modifications for Correspondence (Reviewer ScEQ, Dys6):** As the concept of correspondence and its computation are central to our work, we have added a formal definition and a high-level overview of its computation to Section 3.
- **The relevant descriptions in Figure 2 (Reviewer ScEQ):** We have revised Figure 2 and its caption to clearly distinguish between relational edges (solid lines) and groups of point-to-point edges (dotted lines).
- **The comparisons to LLM-based method and VLA:** we have added a new comparison between our method and relevant LLM-based and VLA approaches.
- **Model Details (Reviewer ScEQ):** We added details on the use of DINO features, data generation statistics (52k pseudo-task groups)

The revised version has been uploaded, and all changes have been highlighted in blue.

**References**

[1] Keypoint Action Tokens Enable In-Context Imitation Learning in Robotics, RSS 2024

[2] RDT-1B: a Diffusion Foundation Model for Bimanual Manipulation, ICLR 2025

[3] Instant Policy: In-Context Imitation Learning via Graph Diffusion, ICLR 2025

[4] DINOv2: Learning Robust Visual Features without Supervision, TMLR

[5] DENSE SEMANTIC MATCHING WITH VGGT PRIOR, Arxiv:2509

---

### Author Response · Authors · 2025-12-03
**Summary of Response to Main Concerns**

We extend our sincere gratitude to the reviewers, Area Chairs and Senior Area Chairs  for their invaluable time and effort.

During the previous rebuttal stage, we were fortunate to receive constructive and actionable reviews from all reviewers. Our work explores in-context policy learning for robotics. Reviewers broadly recognized the **importance** of the topic, noting **excitement** around its connection to meta learning, and the **novelty** of our methodology [iphX, ejwA, ScEQ, Dys6]. They also agreed that our **motivation** and **ideas** were **clearly presented** [ScEQ, Dys6].

Furthermore, our comprehensive experimental evaluation, spanning both simulation and real-world settings, was consistently praised. Reviewers described the results as **"interesting," "promising,"** and **"strong"** [ejwA, ScEQ, iphX, Dys6].

In our rebuttal, we mainly address following primary concerns raised by the reviewers:

1. **Lack of Comparisons:** Our method demonstrates a significant performance improvement over **additional representative VLA and LLM-based counterparts**. These results directly address the reviewers' concerns [ejwA, iphX, ScEQ, Dys6] regarding comparative evaluation.
2. **The Correspondence Extraction Module:** The **[visualized comparison](https://matchingpolicy.github.io/)** between our method and the naive baseline demonstrates its clear advantage and necessity. Furthermore, following the suggestion of Reviewer [Dys6], our approach also outperforms an enhanced baseline that integrates additional geometric features.
3. **Generalization to Real-world Tasks:** We demonstrate **effective one-shot sim-to-real** **transfer**: our method, trained on simulated primitives, adapts to challenging real-world tasks via vision foundation models. Further details  are provided in our response to Reviewer [Dys6].
4. **Method Extensibility:** we provide a detailed discussion to extend our method  [iphX, Dys6] to **multi-objects setting, long-horizon tasks and general single-camera scenarios,** supported by experiments and references to justify our perspective and outline potential future directions.
5. **Paper Clarity:** we have revised the [manuscript](https://openreview.net/pdf?id=ddxq6ACoY3) to provide clearer details on the model input/structure [ScEQ], restructure the Method section [Dys6], and expand the Related Works section [ejwA, Dys6].

We sincerely appreciate all the reviews we received. As the discussion phase was unexpectedly interrupted, we were unable to engage further with most reviewers. Consequently, we are especially grateful for the timely and constructive response from Reviewer [iphx]. They raised their score and provided thoughtful points for discussion, all of which we have addressed through additional experiments and detailed explanations.

Thank you once again for your effort, which has been instrumental in advancing our research.

Authors of Submission2902

---

### Meta-Review · Area_Chair_D1KS · 2026-01-07

**Summary:**

The paper proposes MatchingPolicy, a framework for in-context imitation learning that explicitly decouples visual correspondence extraction from policy execution. By establishing dense semantic correspondences between demonstration and test scenes, the method aims to generalize manipulation skills to unseen objects and layouts.

Despite a responsive rebuttal that introduced significant new baselines, the submission received a borderline reception. While reviewers acknowledged the potential of the correspondence-oriented approach, fundamental concerns regarding hardware constraints and the limits of the in-context learning formulation persisted. Specifically, the method's reliance on a specific multi-camera setup to resolve pose ambiguity was seen as a limitation to broad applicability compared to single-view approaches. Furthermore, the result that performance saturates at 1-2 demonstrations suggesting the method operates more as a one-shot transfer mechanism rather than a scalable learning paradigm. Consequently, despite the empirical improvements over baselines, these limitations prevent it from meeting the bar for acceptance.

**Reviewer Concerns:**

The review process identified several critical issues ranging from experimental comparisons to fundamental hardware and methodological limitations.

- Missing baselines (ejwA, iphX, ScEQ): Initial reviews criticized the lack of comparison against relevant state-of-the-art methods, specifically LLM-based in-context learners and VLA models, to justify the proposed method's performance.
- Multi-camera setting (iphX, Dys6): A significant practical objection was the method's reliance on a specific multi-camera setup to resolve pose ambiguity. Reviewers noted this hardware requirement severely limits the method's applicability "in the wild" compared to more flexible single-view policies.
- In-context learning (iphX): The reviewer questioned whether the method truly benefits from in-context examples in the way LLMs do. The observation that performance saturates at N=1 or 2 demonstrations suggests the method does not scale with more data, undermining the "In-Context Learning" framing.
- Task complexity (Dys6): Concerns were raised about the reliance on synthetic training data generated with privileged information and whether the keypoint-based abstraction could handle complex, long-horizon tasks beyond simple primitives.


The authors successfully addressed the missing baselines  and the confusion regarding the specific mechanics of the correspondence module and graph structure. However, the 3-camera requirement remains a significant barrier. While the authors argued that single-camera setups work for simple layouts, the method's robustness relies on multi-view consistency, which reviewers viewed as a restrictive assumption for a generalist robot policy. The issue of limited scalability also persists. The admission that naive stacking of demonstrations (N>2) does not improve performance confirms that the method operates more as a one-shot transfer mechanism than a scalable in-context learner.

**Reviewer Scores:**

Reviewer iphX will change the score from 4 to 6 after acknowledging the strong new baselines but retained concerns about scaling. Reviewers ejwA, ScEQ, Dys6 will unlikely to change. While they appreciated the baselines, their concerns regarding the hardware constraints and the incremental nature of the contribution relative to the complexity remained unresolved.

---

### Decision · Program_Chairs · 2026-01-26

Reject